# The Role of M1 and M2 Myocardial Macrophages in Promoting Proliferation and Healing via Activating Epithelial-to-Mesenchymal Transition

**DOI:** 10.3390/biomedicines11102666

**Published:** 2023-09-28

**Authors:** Shaowei Kang, Bin Wang, Yanan Xie, Xu Cao, Mei Wang

**Affiliations:** 1Department of Cardiology, The Second Hospital of Hebei Medical University, Shijiazhuang 050000, China; kangshaowei86@126.com (S.K.); wangbinmed@hotmail.com (B.W.); xieyanan860313@163.com (Y.X.); 2Center of Endoscopy, Traditional Chinese Medicine Hospital of Shijiazhuang City, Shijiazhuang 050051, China; 18531128959@163.com

**Keywords:** cardiovascular disease, myocardial infarction, M1 macrophages, M2 macrophages, epithelial-to-mesenchymal transition

## Abstract

(1) Background: The activation of sequential processes for the formation of permanent fibrotic tissue following myocardial infarction (MI) is pivotal for optimal healing of heart tissue. M1 and M2 macrophages are known to play essential roles in wound healing by the activation of cardiac fibroblasts after an episode of MI. However, the molecular and cellular mechanisms mediated by these macrophages in cellular proliferation, fibrosis, and wound healing remain unclear. (2) Methods: In the present study, we aimed to explore the mechanisms by which M1 and M2 macrophages contribute to cellular proliferation, fibrosis, and wound healing. Using both in vivo and cellular models, we examined the remodeling effects of M1 and M2 macrophages on infarcted cardiac fibroblasts and their role in promoting cardiac healing post-MI. (3) Results: Our findings indicate that M1 macrophages induce a proliferative effect on infarcted cardiac fibroblasts by exerting an anti-apoptotic effect, thereby preventing cell death. Moreover, M1 macrophages were found to activate the mechanism of epithelial-to-mesenchymal transition (EMT), resulting in wound healing and inducing the fibrotic process. The present findings suggest that M1 macrophages play a crucial role in promoting cardiac remodeling post-MI, as they activate the EMT pathway and contribute to increased collagen production and fibrotic changes. (4) Conclusions: The present study provides insights into molecular and cellular mechanisms mediated by M1 and M2 macrophages in cellular proliferation, fibrosis, and wound healing post-MI. Our findings highlight the critical role of M1 macrophages in promoting cardiac remodeling by activating the EMT pathway. Understanding these mechanisms can potentially result in the development of targeted therapies aimed at enhancing the healing process and improving outcomes following MI.

## 1. Introduction

Cardiovascular disease (CVD) is the leading cause of mortality worldwide, imposing a deep social and economic impact on public health [1]. Various lifestyle conditions like obesity and smoking result in atherogenesis leading to CVD [2,3]. Among CVD, myocardial infarction (MI) is found to be major contributor to mortality [4]. The necrosis of myocardial cells occurs when the blood supply to the heart decreases or ceases, leading to the development of MI [4]. The death of cardiomyocytes results in localized inflammation that leads to fibrosis, cardiac scarring, and adverse remodeling. These inflammatory conditions and cardiac scarring negatively impact myocardium regeneration, thus potentially resulting in heart failure and death [5].

Recent studies have highlighted the role of macrophages as the key pathological mediators for tissue remodeling and fibrosis [6]. Macrophages are key moderators of the innate immune response that eliminates pathogens through the process of recognition and phagocytosis of pathogens [7]. Microphages play an important role in tissue remodeling via interaction with the extracellular matrix (ECM) of the myocardium [8]. There are heterogeneous populations of macrophages that are classified as M1 or M2 macrophages [9]. In this respect, macrophages are highly plastic in nature and depend on surrounding environments. M1 macrophages are also called activated macrophages and secrete pro-inflammatory cytokines (IL-12, IL-6, IL-1, and IL-23) [10]. Conversely, M2 macrophages are activated macrophages that produce elevated levels of anti-inflammatory cytokines like IL-10 and growth factors including VEGF and matrix metalloproteinases (MMPs) [10]. The polarization of these macrophage subtypes depends on important factors like inhibitor of apoptosis (IAP) proteins, which play important role in deciding the differentiation of macrophages into either M1 or M2.

Prior research has highlighted the implication of M1 and M2 macrophages in cardiac remodeling, which may be associated with the initiation of EMT. Previous studies have reported that activation in M1 but not in M2 macrophages contributes to cardiac remodeling after myocardial infarction in rats. Further, the recruitment of activated macrophages significantly induces ventricular remodeling after myocardial infarction. Also, macrophages have been shown to be involved in various signaling pathways and targeted therapy for myocardial infarction. Nevertheless, the mechanism of M1 and M2 myocardial macrophages’ mediated upregulation of EMT in promoting proliferation and healing in myocardial patients has not been fully addressed. Myocardial healing involves inflammation at the damaged site followed by proliferation of cells, extracellular matrix remodeling, and EMT, restoring the scared myocardium. Despite many clinical and molecular studies investigating the processes of MI scar formation and healing, the molecular mechanism of immune-cell-mediated MI scar healing is not well understood. This study investigates the significance and function of macrophages in both healthy and infarcted myocardium, elucidating how these innate immune cells contribute to extracellular matrix (ECM) remodeling and fibrosis.

## 2. Methodology

### 2.1. In Vitro Experimentation

Hunan Fenghui Biotechnology Co., Ltd. (Changsha, Hunan, China) provided the rat cardiac macrophages. These were cultured in Roswell Park Memorial Institute (RPMI) 1640 Medium (Gibco, Grand Island, NY, USA) and supplemented with 10% fetal bovine serum (FBS) (Gibco) containing 1% antibiotic (Gibco, catalogue number 15240096). The cell cultures were maintained in an incubator (Thermo, Waltham, MA, USA) at 37 °C with 5% CO_2_ for optimal growth. The cultures were monitored daily followed by a changing of the medium every two days to maintain their vitality. The cells were harvested after 70–80% confluency.

### 2.2. Polarization of Rat Cardiac Macrophages

The rat cardiac macrophages were cultured in 6-well plates at a seeding concentration of 700,000 cells per well in 10% fetal bovine serum (FBS) supplemented with an RPMI 1640 medium. After reaching 60–70% confluency, the cardiac macrophages were differentiated into M1 macrophages by a 24 h incubation with 100 ng/mL LPS (Beijing Solaibao Technology Co., Ltd., Beijing, China) containing an RPMI 160 medium. The polarization of cardiac macrophage M2 was performed by a 24 h incubation with 20 ng/mL of IL-4 (PeproTech, Cranbury, NJ, USA) containing an RPMI 1640 medium.

### 2.3. Analysis of CD86 and CD206 by Flow Cytometry

After incubation with LPS and IL-4, the M1 and M2 polarized cells were washed with 1 × PBS followed by detachment with 5 mM Ethylenediaminetetraacetic acid disodium salt solution EDTA (Sigma, St. Louis, MO, USA, CAS 139333). Cell counting was performed, and 2 × 10^5^ cells were suspended in 1 mL of 1X PBS. The cells were then centrifuged at 1000 RPM for 5 min at 4 °C, and the resulting pellet was resuspended in 1.5 mL of PBS. This washing step was repeated twice. The pellet was subsequently resuspended in 100 μL of PBS in a FACS tube. Next, the pellet was mixed with 5 μL of primary antibodies (F4/80 [ab237026], CD86 [ab256270], and CD206 [sc-58986 PE]), and was incubated in the dark at 4 °C for 1 h. Thereafter, 1 mL PBS were added to the cell suspension, followed by centrifugation at 1100 RPM for 6 min at 4 °C. The supernatant was discarded, and 400 μL of PBS were added to the FACS tubes. Finally, the cells were analyzed using flow cytometry with a FACS caliburTM; (BD Biosciences, Franklin Lakes, NJ, USA).

### 2.4. Induction of Cellular Model of MI and Co-Culture with M1 & M2 Macrophages

Rat myocardial fibroblasts (Hunan Fenghui Biological Co., Ltd., Changsha, China) at a seeding density of 2 × 10^4^ cells per well were cultured in a 6-well dish (Orange Scientific, Braine-l’Alleud, Belgium) for 24 h in DMEM (Gibco) medium containing 10% FBS (Gibco), 50 U/mL penicillin, and 50 μg/mL streptomycin. In order to generate the in vitro model of myocardial infarction, cultured rat myocardial fibroblasts were damaged by a 1 h incubation with 100 μM hydrogen peroxide (Sigma, H1009, CAS 7722841) for 24 h, as previously described in the protocol [11]. After incubation with hydrogen peroxide, the damaged fibroblasts were transferred to a fresh plate and maintained in DMEM supplemented with 10% FBS, 50 U/mL penicillin, and 50 μg/mL streptomycin. This medium was used to ensure optimal survival and proliferation of the cells. For co-culture, polarized M1 and M2 macrophages (2 × 104) were seeded on polycarbonate membrane inserts (0.4 μm) and were set into the wells containing the control rat myocardial fibroblasts and MI-induced myocardial fibroblasts for 24 h. This setup ensured that the macrophages and myocardial fibroblasts were in close contact, allowing for optimal interaction between them and the exchange of cellular components. After 24 h, the co-culture was harvested for further analysis. 

### 2.5. Western Blotting

After the treatment, the control myocardial fibroblasts, the MI-induced myocardial fibroblasts, and the M1 and M2 macrophage-treated myocardial fibroblasts were subjected to protein extraction using a RIPA buffer (1000 μL/well) (Beijing Solabel Technology Co., Ltd., Beijing, China) containing 1% PMS F solution (Cat. NO. P0100) (Wuhan Bode Biotechnology Co., Ltd., Wuhan, China). The cell lysates were homogenized, followed by centrifugation at 12,000 rpm for 30 min at 4 °C to remove cell debris. The resulting supernatant was transferred to fresh tubes, and protein concentrations were determined using a Bradford Protein Concentration Determination Kit (AR0145) (Wuhan Bode Biotechnology Co., Ltd.). Subsequently, the proteins were separated by 10% SDS-PAGE followed by transference on PVDF membranes (Cat. NO. IPVH00010) (Millipore, Burlington, MA, USA). After blocking with 5% non-fat dry milk in PBS for 2 h at RT, the membranes were incubated with primary antibodies: TGF-β antibody (1:1000) (Abcam, Cambridge, UK, (ab179695)), Col3a1 antibody (1:1000) (Abcam, (ab7778)), galectin-3 antibody (1:1000) (Abcam, (ab76245)), N-cadherin antibody (1:1000) (Abcam, (ab18203)), vimentin antibody (1:1000) (Cell Signaling Technology (#46173)), E-cadherin antibody (1:1000) (Cell Signaling Technology, (#14472)), and GAPDH antibody (1:1000) (Abcam, (ab9485)) overnight at 4 °C. Following TBST washing, the membranes were incubated with HRP-labeled goat anti-rabbit IgG secondary antibody (ab7090) (1:8000, Abcam) at room temperature for 1.5 h. Chemiluminescence was detected using Western LightningTM Chemiluminescence Reagent (NEL10300EA, Perkin Elmer, Waltham, MA, USA) for 30 s, and the resulting film was scanned and imaged with the Epson Perfection V39 scanner (Epson, Nagano, Japan) to analyze the brightness value of each Western blot band. The results were quantified using ImageJ, and the protein ratio was normalized against GAPDH. No negative or positive controls were included in the study.

### 2.6. Immunofluorescence

Cell fixation was performed using 4% paraformaldehyde in PBS at RT for 5 min. Except for the cells intended for surface antigen staining, the remaining cells were treated with PBS containing 0.1% Triton X-100 at room temperature for 5 min. To block nonspecific antibody binding sites, the cells were preincubated with PBS containing 5% goat serum at room temperature for 30 min. Primary antibodies (vimentin, Galectin-3, N-Cadherin, and E-Cadherin) were then applied to the cells and incubated at room temperature for 1 h. The primary antibody dilutions used were as follows: vimentin (1:1000, Abcam, (ab92547)), galectin-3 (1:1000, Abcam (ab227249)), N-cadherin (1:1000, Abcam, (ab18203)), and E-cadherin (1:1000, Abcam (ab231303)). After washing, the cells were incubated with goat anti-rabbit IgG H&L secondary antibody at room temperature for 1 h, followed by incubation with an anti-fluorescence attenuation mounting agent. No negative or positive controls were included in the study. 

### 2.7. Annexin V-FITC/PI Apoptosis Detection Assay

The induction of apoptosis was studied by an Annexin V-FITC Apoptosis Detection Kit (BD Pharmingen). Approximately 3 × 10^6^ cells/well were seeded in 6-well plates (*n* = 5) and were incubated until 80% confluency at 37 °C and 5% CO_2_. The medium was removed, and the wells were washed with DPBS. The complete medium was added to each well and incubated. One separate plate was untreated. The treated and untreated (control) cells were harvested by centrifugation (1200 rpm for 3 min) separately and resuspended in 1X binding buffer. The cells were stained with 1 µL Annexin V-FITC and 1 µL Propidium Iodide (PI) solution and were incubated for 10 min at room temperature in the dark. Induction of cell death was measured using a flow cytometer (BD CANTO).

### 2.8. MTT Assay

The cytotoxicity effects were analyzed using MTT assay (Himedia). The cells were seeded in 96-well plates (*n* = 4) at a density of 7000–8000 cells per well and incubated at 37 °C and 5% CO_2_ until 100% confluency and were treated with cardiac macrophages M1 and M2. The cells were washed with DPBS and replenished with fresh complete medium. The plates were incubated for 24 h at 37 °C and 5% CO_2_. The freshly prepared MTT reagent was added to the treatment well and incubated for 4 h. Thereafter, DMSO (100 µL) was added to each well and incubated at room temperature for 10 min. The optical density of the formazan crystal formed in each well was measured at 570 nm with a reference at 630 nm using an ELISA plate reader (ALTA). 

### 2.9. Experimental MI Model

The research experiments involving animals were granted ethical approval by the Hebei Medical University (protocol code: HB2H202211) and conducted in accordance with all applicable guidelines, regulations, and ethical standards. Male wistar rats weighing 200 ± 20 g were used in the study. The rats were either subjected to a permanent ligation of the left anterior descending (LAD) artery or underwent a sham operation without ligation, following previously described methods [12]. After that, the rats were anesthetized with 10% chloral hydrate (0.3 mL/100 g), intubated, and ventilated using a rodent respirator. A left thoracotomy was performed to expose the heart, allowing visualization of the LAD artery, which was ligated permanently using a 4–0 silk suture at the site where it emerges beneath the left atrium. The sham-operated rats underwent the procedure without LAD ligation. After the surgery, the rats were randomly divided into a control group, a model group, a model group with M1 macrophages, and a model group with M2 macrophages. The experiment comprised 24 animals, divided into four groups, each containing six animals. The control group received no injections, while in the model group, 100 μL of pure PBS solution was injected at four points between the ischemic area of the left ventricle’s anterior wall and normal myocardial tissue. The M1 macrophage group received an injection of 100 μL PBS solution containing M1 macrophages (2 × 10^9^ cells/mL), and the M2 macrophage group received the same volume of PBS solution containing M2 macrophages (2 × 10^9^ cells/mL). Following surgery, each rat received a daily intraperitoneal injection of 10,000 U of penicillin for infection prevention over five consecutive days. After an additional 24 h period, the rats were anesthetized and euthanized, and their heart tissues were collected for analysis. The hearts were promptly frozen in liquid nitrogen for subsequent studies.

### 2.10. Masson Staining

The fixation of the heart was performed in 4% buffered paraformaldehyde at pH 7.0 and kept at room temperature for 24 h. Further, the tissue was embedded, sectioned into 4 µm slices, and stained with a Masson’s trichrome. The samples were observed using light microscopy (Olympus BX60, Beijing, China). To calculate the collagen area, ten random microscopic fields were selected from each left ventricle (LV) slide, magnified at ×400.

### 2.11. Immunocytochemistry

The fixation of tissue sections was performed with 4% paraformaldehyde in PBS for 5 min at room temperature. Following fixation, the sections were washed at room temperature with PBS containing 0.1% Triton X-100 for 5 min. To block nonspecific antibody binding, the sections were pretreated with PBS containing 5% goat serum for 30 min at room temperature. Subsequently, primary antibodies (vimentin, galectin-3, N-cadherin, and E-cadherin) were applied to the slides and incubated at room temperature for 1 h. The primary antibody dilutions used were as follows: vimentin (1:1000, Abcam), galectin-3 (1:1000, Abcam), N-cadherin (1:1000, Abcam), and E-cadherin (1:1000, Abcam). After rinsing, the sections were incubated with HRP-labelled goat anti-rabbit IgG secondary antibody (ab6721) for 1 h at room temperature.

### 2.12. TUNEL/DAPI Immunofluorescence

Paraffin-embedded sections of the left ventricle, encompassing both non-infarcted and infarcted areas, were subjected to dewaxing and dehydration. Protease K (20 μg/mL dissolved in Tris/HCl, pH 7.4~8.0) was then applied, and the sections were incubated at room temperature for 15–30 min. After two washes with PBS, the sections were treated with 50 μL of TUNEL reaction mixed solution in a wet box at 37 °C for 1 h. To prevent evaporation and ensure uniform distribution of the TUNEL reaction mixture, the glass slides were covered. TUNEL-positive staining was subsequently examined under a fluorescence microscope after three washes with PBS. Following this, the sections were incubated with 50 μL of conversion agent-POD in a wet box at 37 °C for 30 min. Finally, after three washes with PBS, the sections were incubated with 50–100 μL of DAPI solution at room temperature for 5 min. 

### 2.13. Statistics

Statistical analyses were conducted using GraphPad Prism software, version 8 (GraphPad Software Inc., San Diego, CA, USA). The data are presented as mean ± SEM. For comparisons among multiple groups, repeated-measures analysis of variance (ANOVA) or 1- or 2-way ANOVA was performed. Two-group comparisons were evaluated using the two-tailed, unpaired Student’s *t*-test.

## 3. Results

### 3.1. Polarization of Rat Cardiac Macrophages into M1 and M2 Populations

The rat cardiac macrophages were differentiated into M1 and M2 macrophages by incubation in the presence of LPS and IL-4. The expression of CD 86 (Cluster of differentiation 86) and CD 206 (Cluster of differentiation 206) were analyzed using flow cytometry to characterize the efficacy of the M1/M2 polarization. The untreated cardiac macrophages’ M0 cells were used to demonstrate the basic gating strategy of flow cytometry. The flow cytometry analysis measured the increased expression of CD86 (74.7%) expressing M1 polarized macrophages, whereas the M2 polarized macrophages had an increased expression of CD 206 (70%) as compared to the undifferentiated M0 cardiac macrophages. Figure 1 depicts the comparative dot plot of CD86 and CD206 expressing undifferentiated, M1 and M2 polarized macrophages.

### 3.2. Validation of In Vitro MI Model and Activation of Fibrosis in Cardiac Fibroblast

We induced the H_2_O_2_-mediated apoptosis in cardiac fibroblasts that corresponds to MI-like cellular pathology. The M1 and M2 cardiac macrophages were co-cultured with the MI-induced cells to study the expression of TGF-β 1 and Col3a1. Following H_2_O_2,_ the protein expression of TGF-β 1 and Col3a1 was found to be significantly increased 2.9-fold (*p* < 0.001) and 2.7-fold (*p* < 0.001) in the H_2_O_2_-treated cells as compared to the control cardiac fibroblast. Similarly, the co-culturing with the M1 macrophages further elevated the protein expression of TGF-β 1 and Col3a1 1.25-fold (*p* < 0.01) and 1.4-fold (*p* < 0.001) as compared to the MI model group. However, the co-culturing with the M2 macrophages downregulated the protein expression of TGF-β 1 and Col3a1 1.25-fold (*p* < 0.01) and 1.4-fold (*p* < 0.001) as compared to the MI model group (Figure 2A,B). The upregulation of TGF-β 1 and Col3a1 in the MI model group suggests the activation of cardiac fibrosis. 

### 3.3. Estimation of Cellular Proliferation and Apoptosis after Treatment of M1 and M2 Macrophages

We estimated the cellular proliferation in the control cardiac fibroblast, MI model, M1, and M2 co-cultured cardiac fibroblast at different time durations: 0 h, 24 h, and 48 h. The mean optical density calculated in the MTT assay demonstrated different growth responses at different time durations. After 24 h, we observed an increased cellular proliferation in the MI-induced model (mean ± S.D; 0.725 ± 0.02) and the M1 co-cultured fibroblast (0.89 ± 0.031) as compared to the control group. By contrast, the M2 co-cultured cardiac fibroblast showed decreased cellular proliferation (0.62 ± 0.02) as compared to the MI-induced cardiac fibroblast. After 48 h, incubation with the M1 macrophages further increased the proliferation of the cardiac fibroblast by a mean± S.D value of 1.17 ± 0.02 as compared to the MI-induced fibroblast. On the other hand, the treatment of the M2 macrophages decreased the proliferation of the cardiac fibroblast by mean ± S.D value of 0.8 ± 0.017. The data are shown in Figure 3A.

In order to study the apoptosis-inducing effect of M1 and M2 macrophages on cardiac fibroblast, we performed the co-culture of the M1 and M2 macrophages with the MI-induced cardiac fibroblast cells. After the 24 h incubation of the fibroblast with M1 and M2 macrophages, annexin V-FITC/PI staining was examined by flow cytometry to investigate the apoptosis rate among the control, the MI-induced model, and the M1- and M2- treated fibroblast cells. As shown in Figure 3B, the cardiac fibroblast treated with M1 macrophages had 6.06 ± 0.8% of apoptosis rate as compared to the MI-induced model (13.38 ± 0.6%) and the M2-treated cardiac fibroblast (18.1 ± 0.8%). This illustrates that treatment of cardiac fibroblast with macrophages significantly decreases the apoptosis rate and thereby increases cellular proliferation. 

### 3.4. M1 Macrophages Promote the Activation of Epithelial-to-Mesenchymal Transition in Humans

To determine whether treatment of MI-induced cardiac fibroblast with M1 and M2 macrophages induces the activation of EMT in fibroblast, we performed the immunofluorescence staining of galectin-3, N-cadherin, vimentin, and E-cadherin in each group. The results indicated that increased localization and expression of galectin-3, N-cadherin, and vimentin (Figure 4A–C) was found in the cardiac fibroblast treated with the M1 macrophages as compared to the control, the MI model, and the M2-treated fibroblast cells. However, the expression of E-cadherin was found to be increased in MI-induced cardiac fibroblast treated with the M2 macrophages.

Furthermore, we also performed Western blot analysis to quantitate the protein expression of galectin-3, N-cadherin, vimentin, and E-cadherin in each group. The results indicated that the relative protein expression of galectin 3, N-cadherin, and vimentin was found to be significantly elevated by 1.38, 1.21, and 1.27 as compared to the MI-induced cardiac fibroblast. On the other hand, the expression of E-cadherin decreased in the MI-induced cardiac fibroblast treated with M1 macrophages but was found to be significantly upregulated in the M2-treated MI-induced cardiac fibroblast by 1.36. The data are shown in Figure 5A–D. Taken together, the findings of the immunofluorescence and Western blot analysis illustrate that M1 macrophages promote the activation of ETM via upregulation of galectin-3, N-cadherin, and vimentin and via downregulation of the expression of E-cadherin.

### 3.5. In Vivo Evaluation of Collagen Synthesis and Fibrotic Changes

We evaluated the fibrotic lesions and type 1 collagen deposition in cardiac tissue of the control, the MI-induced model, and the M1- and M2-treated MI groups. Masson’s trichrome staining was performed among these groups to assess the deposition of type 1 collagen in the cardiac tissue. Masson trichome staining depicts the increased deposition of collagen 1 in the M1 macrophages-treated MI group; however, we observed less deposition of collagen 1 in the M2-treated MI group (Figure 6). In addition, cardiac fibrosis and structural observations were also confirmed by Haematoxylin-eosin staining (Figure 7) H & E depicts increased fibrosis in the M1 macrophages-treated MI group. 

### 3.6. M1 Macrophages Promote the Activation of Epithelial-to-Mesenchymal Transition in Infarcted Rat Myocardium Treated with M1 Macrophages 

To determine whether the mechanism of EMT is activated in MI-induced and M1 & M2 macrophages–treated MI rats, we performed an immunohistochemistry of galectin-3, N-cadherin, vimentin, and E-cadherin for each group. The results indicated that the expression of galectin-3, N-cadherin, and vimentin (Figure 8A–C) was found to increase in cardiac tissue sections treated with M1 macrophages as compared to the control, the MI model, and the M2-treated model. However, the expression of E-cadherin (Figure 8D) was found to be increased in the cardiac tissue sections of the M2 macrophage–treated animal model. The results were confirmed by Western blot analysis of galectin-3, N-cadherin, vimentin, and E-cadherin. Similar to the findings, the expression of galectin-3, N-cadherin, and vimentin was found to be significantly upregulated in the cardiac tissue sections treated with M1 macrophages as compared to the control, the MI model, and the M2- treated model. By contrast, the protein expression of E-cadherin was found to be increased in the cardiac tissue sections of the M2 macrophage–treated animal model. The data are shown in Figure 9A–D.

### 3.7. Evaluation of DNA Fragmentation–Based Apoptosis

We performed the TUNEL assay (terminal deoxynucleotidyl transferase-mediated d-UTP nick end labeling) in paraffin-embedded left ventricle heart sections among different groups. As shown in Figure 10, the increased TUNEL-positive cells were detected in the cardiac tissue sections treated with the M1 macrophages and in the MI model.

## 4. Discussion

Cardiovascular disorder (CVD) is the leading global cause of illness and death and encompasses conditions like hypertension, coronary heart disease, and myocardial infarction. In all these conditions, inflammation plays a dual role, both as a contributor and as an exacerbating factor in CVD, impacting its prognosis [13]. In this respect, myocardial infarction occurs due to cardiomyocyte death, leading to activation of the complex innate immune system and resulting in the induction of a series of inflammatory responses, which is a requisite for remodeling, healing, and scar formation [14]. In this study, we described the molecular mechanism of MI scar wound healing and fibrotic changes mediated by different populations of macrophages.

In this study, we evaluated the mechanistic role of M1 and M2 rat cardiac macrophages on cellular myocardial infarcted fibroblast and in the MI rat model. Their major subtypes of macrophages were classified into M1 and M2 based on cell surface receptors [15,16]. Our results indicated that the polarization of M1 and M2 macrophages was consistent with previous reports [15]. In vitro evaluation demonstrated the proliferative and antiapoptotic effect mediated by M1 macrophages in cardiac fibroblast, whereas the M2 macrophages exerted an apoptotic effect on MI-induced cardiac fibroblast. This might be due to the expression of BMP-2 proteins by M2 macrophages after co-culture with cardiac fibroblast, as previous studies have demonstrated that BMP proteins, especially BMP2 expressed by M2 macrophages, induce apoptosis [17]. Many reports related to oral cell carcinoma reported consistent findings on the proliferative effect of M1 macrophages [18]. Few studies depict that M1 macrophages secrete proinflammatory cytokines that can contribute to inflammation [19].

In addition to providing detailed evidence that cardiac macrophages are important determinants of cardiac remodeling in the context of myocardial infarction, the present study illustrates the importance of carefully categorized M1 and M2 macrophage populations to identify specific subsets that might serve as a therapeutic target to heal the infarct tissue. In this respect, we have also investigated the molecular mechanism of EMT that is mediated by M1 and M2 macrophages. The term “epithelial-to-mesenchymal transition” (EMT) refers to a sequence of processes in which polarized epithelial cells attach to the basement membrane and undergo various molecular changes to transform into mesenchymal-like cells. EMT is a physiological process that plays an important role in organogenesis, tissue development, and wound healing [20]. Many studies have highlighted the activation of scar formation after MI [12]. In our study, we found that M1 macrophages induce the process of EMT in MI-induced cardiac fibroblasts by upregulating the protein expression of galectin-3, N-cadherin, and vimentin and downregulating E-cadherin expression, whereas M2 macrophages upregulate E-cadherin expression and downregulate the expression of galectin-3, N-cadherin, and vimentin. We found consistent results in the MI-induced animal models also. Previous literature reported that galectin-3, N-cadherin, and vimentin are important mediators of EMT, especially in multiple tumors [21].

Furthermore, this is the first study to report the activation of EMT in MI-induced cardiac fibroblast by M1-polarized macrophages. The activation of EMT by M1 macrophages may represent an indispensable step for wound healing and scar remodeling. As per the previous clinical, molecular, and histological studies, the synthesis of collagen type I after myocardial infarction represents an important step in scar healing [22]. Furthermore, the progression of myocardial wound healing following infarction also involves fibrosis and long-term maturation [23]. The increased deposition of type 1 collagen and increased fibrosis were significantly upregulated by treatment with M1 macrophages. This effect demonstrated the scar healing properties mediated by M1 macrophages in cardiac fibroblast following MI-like injuries. Various studies have reported myofibril healing after MI [24,25,26]. Also, very few studies have reported the role of M1 macrophages in skin regeneration during tissue expansion [27,28,29]. Previous studies have also reported the role of resident macrophages in the process of wound healing following an episode of MI. So, the contribution of M1/M2 macrophages represents the initiation of an EMT process that plays an important role in MI wound healing.

## 5. Conclusions

The present findings represent the first study to provide evidence that M1-polarized macrophages play an essential role via the activating pathway of EMT in the promotion of cardiac remodeling in post-MI rats as well in an in vitro model. The results suggest that a high M1/M2 macrophage ratio is a potential target for intervention for wound healing after MI.

## Figures and Tables

**Figure 1 biomedicines-11-02666-f001:**
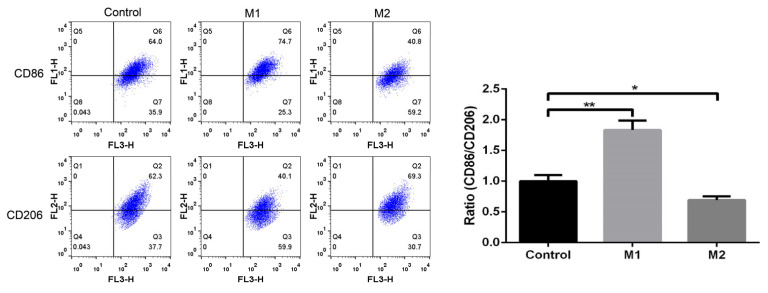
Polarization of rat cardiac macrophages: the expression of CD86 and CD206 on M1 and M2 polarized rat cardiac macrophages (* *p* < 0.05, ** *p* < 0.01).

**Figure 2 biomedicines-11-02666-f002:**
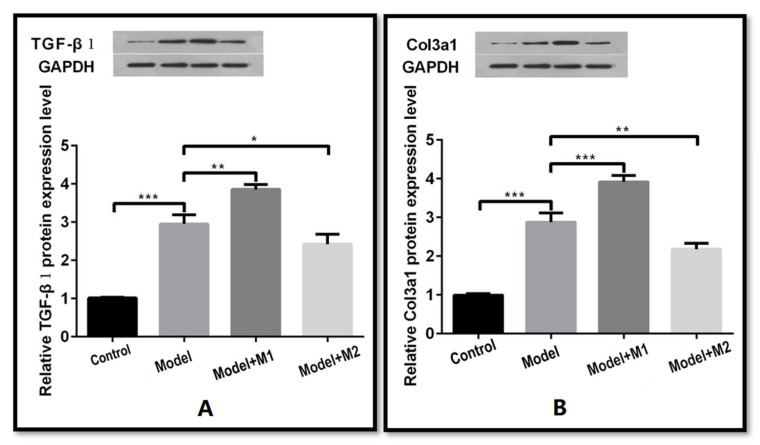
(**A**,**B**) Western blot analysis of TGF-BETA and Col3a1 (* *p* < 0 05, ** *p* < 0 01, *** *p* < 0.001).

**Figure 3 biomedicines-11-02666-f003:**
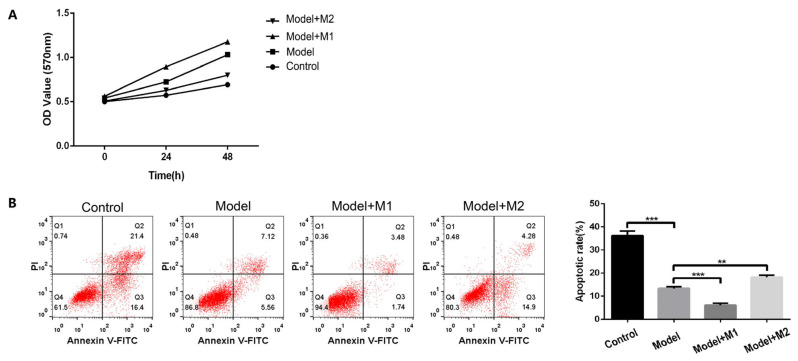
(**A**) Proliferationof cells in differentgroups (**B**) Apoptosis in each group by flow cytometry (** *p* < 0 01, *** *p* < 0.001).

**Figure 4 biomedicines-11-02666-f004:**
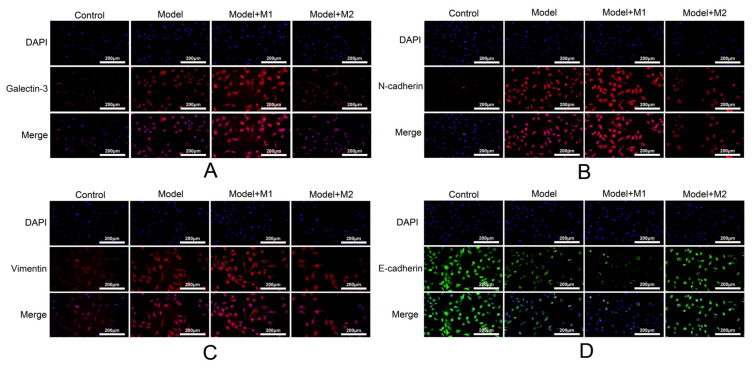
(**A**) Expression of Galectin-3 was detected in each group; (**B**) expression of N-cadherin; (**C**) expression of vimentin; (**D**) expression of E-cadherin.

**Figure 5 biomedicines-11-02666-f005:**
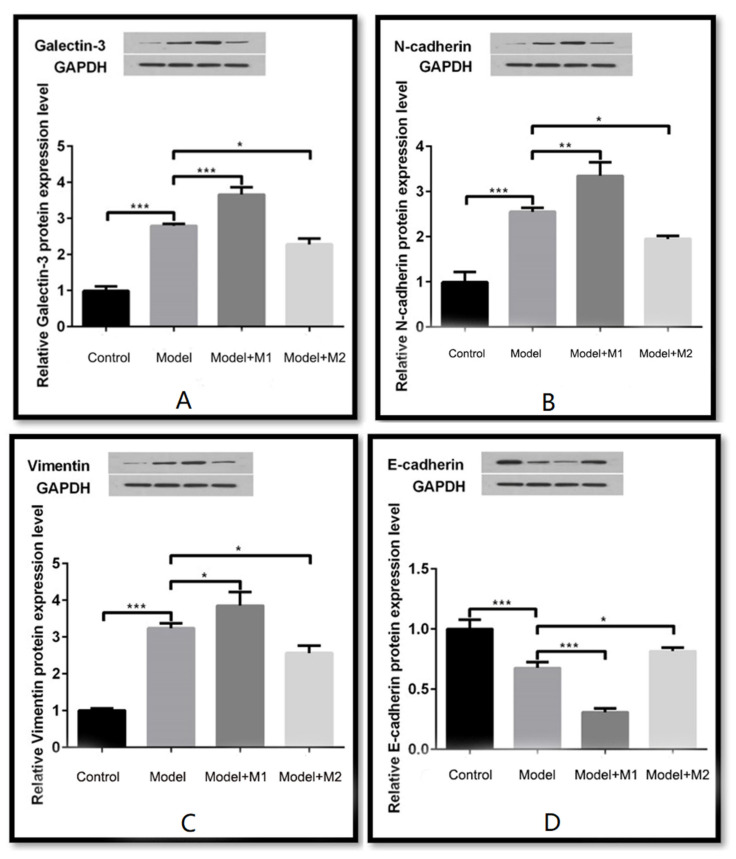
(**A**) Western blot detected the expression level of its gene galectin-3; (**B**) Western blot detected the expression of N-cadherin; (**C**) Western blot detected the expression of vimentin; (**D**) Western blot detected the expression of E-cadherin. (* *p* < 0.05, ** *p* < 0.01, *** *p* < 0.001).

**Figure 6 biomedicines-11-02666-f006:**
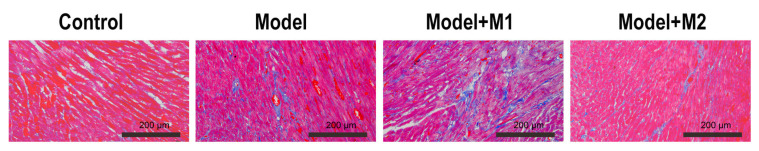
Masson’s trichrome staining to demonstrate the deposition of type 1 collagen in myocardium tissue.

**Figure 7 biomedicines-11-02666-f007:**
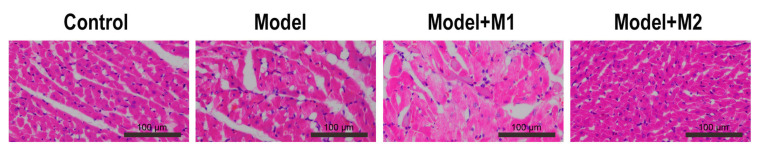
HE staining detected the fibrosis in different treatment groups.

**Figure 8 biomedicines-11-02666-f008:**
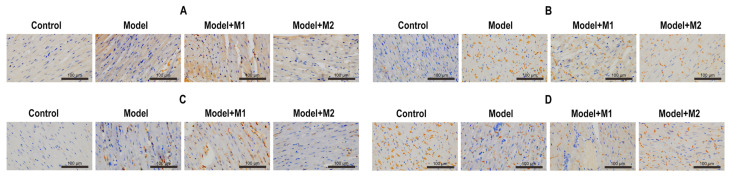
(**A**) The expression of galectin-3 was detected in each group; (**B**) the expression of N-cadherin; (**C**) the expression of vimentin; (**D**) the expression of E-cadherin.

**Figure 9 biomedicines-11-02666-f009:**
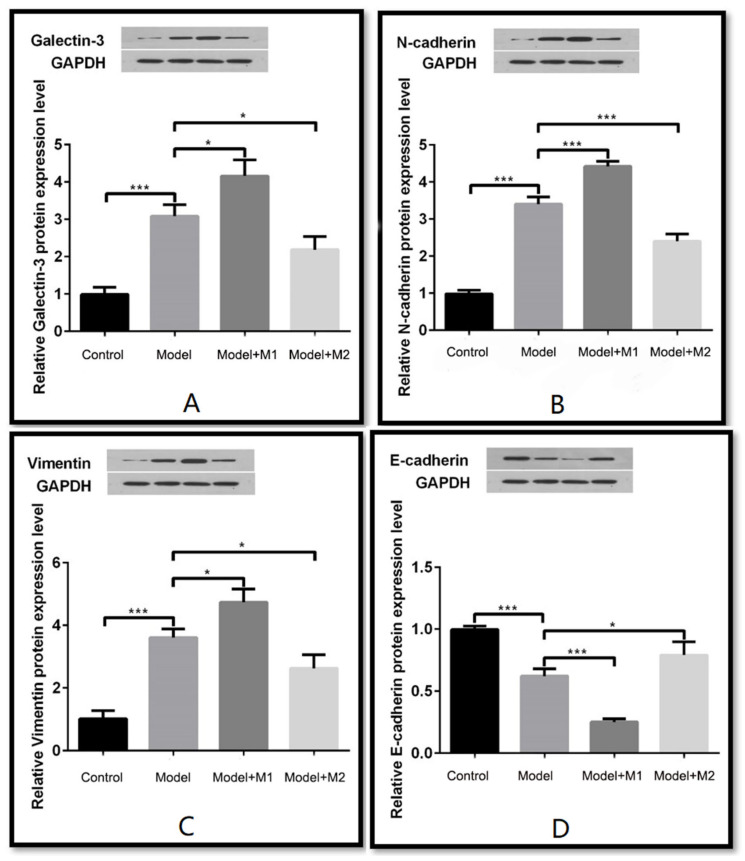
(**A**) Western blot of cardiac rat tissue detected the expression level of its gene galectin-3; (**B**) Western blot detected the expression of N-cadherin; (**C**) Western blot detected the expression of vimentin; (**D**) Western blot detected the expression of E-cadherin. (* *p* < 0.05, *** *p* < 0.001).

**Figure 10 biomedicines-11-02666-f010:**
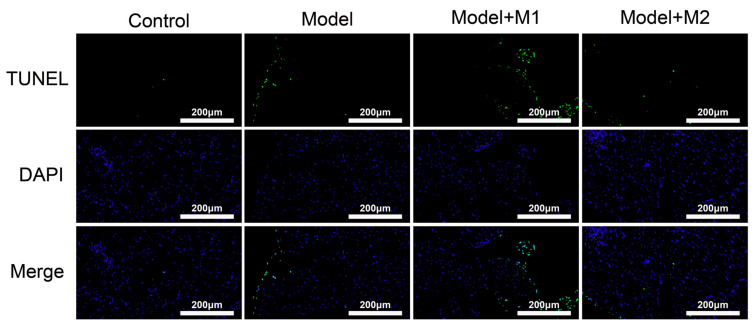
TUNEL detected apoptosis in different treatment groups.

## Data Availability

The initial structures and the resulting data and simulations are available in this manuscript only.

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
