# Peer review of "The Role of M1 and M2 Myocardial Macrophages in Promoting Proliferation and Healing via Activating Epithelial-to-Mesenchymal Transition"

_biomedicines, 2023, doi:10.3390/biomedicines11102666_

Round 1
Reviewer 1 Report
The manuscript (MS) entitled “Role of M1 and M2 myocardial macrophages in promoting proliferation and healing via activating epithelial to mesenchymal transition” by Shaowei Kang and colleagues presents experimental findings about the role of M1 and M2 macrophages in cell proliferation, fibrosis, and wound healing in the post myocardial infarction (MI) milieu in the in vitro and in vivo models of MI.
I find the results interesting and worth publishing if the authors significantly improve description of methods and results.
My main concern is related to methods, which lack significant pieces of information. Specifically, it is not clear how many animals there are per each group/time point, how macrophage treatment was implemented, how many tissue samples were used for imaging and Western blotting. Please, see my detailed comments below.
Major points:
1. Please, clarify in the MS how many animals were used for experiments, how many are included in each group, how many time points of tissue collection there are (and at what period?). This should be also included in the figures (how many rats and samples are used for each evaluation).
2. For imaging data – clarify if this is from a single section/slide, or a representative image of a larger cohort of images. Please, clarify, which parts of the left ventricle were used for evaluation (MI-free? Scar tissue? Border between scar and normal myocardium?), at what time point was the evaluation done? How many hearts were evaluated?
3. Please, specify catalogue numbers of antibodies used in experiments (this could be provided either as a separate table or indicated next to the manufacturer of antibodies).
4. Were there any positive or negative controls performed for primary/secondary antibody specificity for immunofluorecent imaging and for Western blotting? Please, indicate in the MS.
5. The in vivo model of MI and macrophage treatment is not clear. Were M1/M2 macrophages administered into the MI-rats? Please, clarify this protocol and provide specific time points that were used for evaluation of the cardiac tissue.
6. Please, confirm that all data fulfilled ANOVA criteria (normality, sphericity, etc.). For data that does not fulfil ANOVA requirements, corresponding to ANOVA non-parametric tests should be carried out. Such data should be also presented as median value with interquartile range (IQR).
7. Results of macrophages differentiation and Figure 1 – is there any statistical analysis of the results? Figure 1 is not very convincing that macrophages are clearly differentiated into two different populations. I cannot see a clear-cut difference between M1/M2/M0 macrophages (unless by checking the percentage values). Statistical analysis or CD86 vs CD206 plot could be more convincing?
8. Fonts in the figures in the MS are quite small and illegible. Figures should be improved in this regard - especially Fig 3, Fig 4, Fig 5, Fig 8, Fig 9. I see that in the supplementary material the figures are legible, however, they do not seem to fit well in the MS.
9. Results – Fig.10 – it seems that there is no apoptosis in the MI group without any treatment in the version in the MS. The same figure in the supp.material is better (but still quite "empty" and dark). Is it possible to improve Fig 10 in the MS? In addition, please, confirm how many sections were evaluated, which parts of the left ventricle were used for evaluation (MI-free? Scar tissue? Border between scar and normal myocardium?), at what time point was the evaluation done? How many hearts were evaluated?
10. There are more original experimental papers that show involvement of macrophages in cardiac remodelling after MI. I strongly suggest including a broader range of original papers, and not to rely mostly on review papers.
Specifically:
Important relatively recent review in the field:
Peet C, Ivetic A, Bromage DI, Shah AM. Cardiac monocytes and macrophages after myocardial infarction. Cardiovasc Res. 2020 May 1;116(6):1101-1112. doi: 10.1093/cvr/cvz336. PMID: 31841135; PMCID: PMC7177720.
The paper showing the role of regulatory T cells in differentiation of macrophages post-MI:
Weirather J, Hofmann UD, Beyersdorf N, Ramos GC, Vogel B, Frey A, Ertl G, Kerkau T, Frantz S. Foxp3+ CD4+ T cells improve healing after myocardial infarction by modulating monocyte/macrophage differentiation. Circ Res. 2014 Jun 20;115(1):55-67. doi: 10.1161/CIRCRESAHA.115.303895. Epub 2014 Apr 30. PMID: 24786398.
The paper showing that shift from M1 to M2 improves healing after MI:
Liu D, Guo M, Zhou P, Xiao J, Ji X. TSLP promote M2 macrophages polarization and cardiac healing after myocardial infarction. Biochem Biophys Res Commun. 2019 Aug 20;516(2):437-444. doi: 10.1016/j.bbrc.2019.06.041. Epub 2019 Jun 19. PMID: 31227217.
The paper showing the effect of CD226 on macrophage polarization and healing of MI heart:
Li J, Song Y, Jin JY, Li GH, Guo YZ, Yi HY, Zhang JR, Lu YJ, Zhang JL, Li CY, Gao C, Yang L, Fu F, Chen FL, Zhang SM, Jia M, Zheng GX, Pei JM, Chen LH. CD226 deletion improves post-infarction healing via modulating macrophage polarization in mice. Theranostics. 2020 Jan 20;10(5):2422-2435. doi: 10.7150/thno.37106. PMID: 32104514; PMCID: PMC7019150.
(please, note I have not authored any of the above papers, I am not associated with any of the authors nor with any of the journals the papers were published in).
Minor points:
1. The MS will benefit from careful proofreading and some corrections of typographical/grammatical errors.
Please, see some examples (many more in the MS):
“Understanding these mechanisms can potentially results in development” should be “…mechanisms can potentially result in…” (L28-29)
“the mechanism of M1 and M2 myocardial macrophages mediated upregulation of epithelial to mesenchymal transition in promoting proliferation and healing to myocardial have not been fully addressed” (L55-56) - please, clarify “healing to myocardial…”
“Hunan Fenghui Biotechnology Co., Ltd. (Changsha, Hunan, China) have provided” (L63) – have or has?
“Figure 3. a Proliferationof cells” (L263) – “a” should be removed.
2. Please, confirm that “sheep anti-rabbit IgG secondary antibody (ab7090) (1:8000, Abcam)” (L121-122) is a sheep anti-body, as per Abcam website, the ab7090 antibody is a goat anti-rabbit.
3. “the LAD artery, which was ligated permanently using a 4-0 silk suture at the site where it emerges from the left atrium. (L168-169) – I suggest to change the sentence to “…from beneath the left atrium”, as LAD does not emerge from the atrium.
4. Microscopic images are of different sizes, even within the same figure. Please, streamline this.
5. Formatting of references should be streamlined.
The English language is generally good, but it could be improved.
Please, note, I am not a native speaker.
Author Response
Kindly check the attachment.

Reviewer 2 Report
I reviewed with interest the manuscript by Shaowei Kang et al. "Role of M1 and M2 myocardial macrophages in promoting proliferation and healing via activating epithelial to mesenchymal transition". In this article, the authors studied the mechanism of myocardial macrophage M1 and M2 mediated activation of the epithelial-mesenchymal transition in promoting myocardial proliferation and healing. They showed that polarized M1 macrophages play an important role by activating the epithelial-mesenchymal transition pathway in stimulating cardiac remodeling in post-MI rats. This is interesting new data, but during the review I had comments and questions.
1. In my opinion, the authors in the introduction did not sufficiently reflect the novelty of the study. On lines 53-57 the authors note: "Prior research has highlighted the implication of M1 and M2 macrophages in cardiac remodeling, which may be associated with the initiation of epithelial to mesenchymal transition. Nevertheless, the mechanism of M1 and M2 myocardial macrophages mediated upregulation of epithelial to mesenchymal transition in promoting proliferation and healing to myocardial have not been fully addressed". However, the authors do not provide any references to previous studies. At the same time, a number of studies on this issue clearly needed to be mentioned. For example, a study by Liu et al showed that Activation in M1 but not M2 Macrophages Contributes to Cardiac Remodeling after Myocardial Infarction in Rats (1). Macrophage differentiation and M1/M2 polarization are discussed in Morón-Calvente et al (2), macrophage involvement in Signaling pathways and targeted therapy for myocardial infarction in Zhang et al (3). There are also publications on the possible impact of therapy on ventricular remodeling after myocardial infarction through inducing alternatively activated macrophages (4). Undoubtedly, the authors should consider such publications both in the introduction and in the discussion of the results obtained.
2. It is better to start the discussion section with the results obtained by the authors, that is, the first paragraph of the discussion should be moved to the introduction.
3. Minor:
- reference format 2 is incorrect, in this case it is also better to provide a link to data from epidemiological studies
- there is no link in the text to link 22
- on line 392 - a typo? (a high M1/MA macrophage ratio)
References:
1. Liu W, Zhang X, Zhao M, Zhang X, Chi J, Liu Y, Lin F, Fu Y, Ma D, Yin X. Activation in M1 but not M2 Macrophages Contributes to Cardiac Remodeling after Myocardial Infarction in Rats: a Critical Role of the Calcium Sensing Receptor/NRLP3 Inflammasome. Cell Physiol Biochem. 2015;35(6):2483-500. doi: 10.1159/000374048.
2. Morón-Calvente V, Romero-Pinedo S, Toribio-Castelló S, Plaza-Díaz J, Abadía-Molina AC, Rojas-Barros DI, Beug ST, LaCasse EC, MacKenzie A, Korneluk R, Abadía-Molina F. Inhibitor of apoptosis proteins, NAIP, cIAP1 and cIAP2 expression during macrophage differentiation and M1/M2 polarization. PLoS One. 2018 Mar 8;13(3):e0193643. doi: 10.1371/journal.pone.0193643.
3. Zhang Q, Wang L, Wang S, Cheng H, Xu L, Pei G, Wang Y, Fu C, Jiang Y, He C, Wei Q. Signaling pathways and targeted therapy for myocardial infarction. Signal Transduct Target Ther. 2022 Mar 10;7(1):78. doi: 10.1038/s41392-022-00925-z.
4. Li J, Shen D, Tang J, Wang Y, Wang B, Xiao Y, Cao C, Shi X, Liu HM, Zhao W, Zhang J. IL33 attenuates ventricular remodeling after myocardial infarction through inducing alternatively activated macrophages ethical standards statement. Eur J Pharmacol. 2019 Jul 5;854:307-319. doi: 10.1016/j.ejphar.2019.04.046.
No comments
Author Response
Kindly check the attachment.

Reviewer 3 Report
Congratulations to Authors! Excellent article, very important topic.
My remarks are from Line 11 - to Line 390 (please read until 390 here below).
Before reading every comment, here is an important foreword:
Minor revision (the most important shown below is that: at Line 265 EMT and Line 306 EMT results seem double shown. These subchapters with immunofluorescence and Western blot are DOUBLE shown. Figure 5 and Figure 9 are nearly the same (with the exception of some significances,asterixes on panels 5b and 9b.Please check and omit duplicates)
Line 11 pertinent permanent
Line 15 remain incomprehensible are less understood
Line 16 In the present
Line 23 play an crucial role (the correct is: play a crucial role)
Line 18 : remodelling Line 28: remodeling please use either with single or double l letter
(remodeling is used later)
Line 28-29 : can potentially result (after „can”, the verb can not be equipped with s)
Line 37: to be the major to be an important (the „major” is reflected for coronary artery disease /CAD/ in the cited article) CVD involves not only CAD but HF, arrhythmias etc.
Line 74: 100 ng/ml (space was missing)
Line 76: 20 ng/ml (space was missing)
Line 79: cells were washed
Line 81: cells were suspended
Line 87: PBS was added
Line 88: 2 dots are at the end of the sentence. : (BD Biosciences)..
Line 92: dot is unnecessary after FBS (Gibco).,
Line 95: 24 h
Line 142: cells/well were seeded (maybe the „were” is missing)
Line 155: 24 h
Line 156: MTT reagent was added
Line 164: 200 ± 20 g were used
Results
3.1
Line 211: Please check whether you would like to use capital letters:
Polarization of Rat cardiac (Cardiac) Macrophages …….
Figure 1. Panel left upper : please correct „Controrl” to „Control”
Line 217: Please check whether the numeric value (75.7%) /M1 CD86/ in the text is the same
with the value shown on the corresponding Figure 1 panel showing 74.7%
Line 219: 70% is written in the results and 69.3% is shown at Panel CD206 M2. Please check.
Line 223: Rat rat (not needed the capital letter)
Line : 226 and Line 230 H2O2 please check to apply H2O2 or H2O2 (subscript)
Line 230 and 231: fibroblasts (plural needed, please check)
Line 233 : downregulated (past tense)
Line 234 and related Figure 2 :
Figure 2 in Line 234 the significances are shown numerically. It is OK, but: the corresponding number of asterixes on Figure 2 at „ Model + M2” bars might require one more asterix in case of TGF-B1 and one more asterix in case of Col3a1. Please check the asterixes.
Line 241: i.e. 0-hour, 24-hour and 48-hour or 0 hour, 24 hour and 48 hour
Line 244 and 244: you can consider typing in the control numeric values also
Figure 3b Maybe a little increase in numeric values could be considered
Please use small „p” instead of capital letter „P” in Figure legends at Figure 3.
Please write in ( if you would like to ) the p values into Figure 2 legend similarly as it is seen at Figure 3 legend.
Lines 248 and 250: you can also use past tense: increased (line 248), decreased (line 250)
Line 257: you can use plural: fibroblasts
Line 258 and 259 : …and M2 treated fibroblasts have 18.1 ……. (verb was needed)
Figure 3: Why the treatment (by H2O2) group „Model” resulted in less apoptotic rate than control?
You might increase numeric data size where needed, for better visibility.
Line 259: with M1 macrophages
Line 263: Figure 3 Legend: you can insert : …in each group by flow cytometry at 24 h
Line 264: „P” could be changed to „p” (not needed capital letter)
Line 275: 4B (instead of 4b). Please make universal through whole manuscript text whether you use capital or normal letter at Figure panels naming.
Line 289 – 291 : Figure 5 A („A” or „a” is missing, please make universal capital letter or not).
Line 291: Asterixes are missing from p<0.001
Figure 5: 5b Font size is a bit smaller (e.g. with 1 magnitude) at x axis : „Control” „Model” „Model +M1”, „Model +M2” in comparison with other panels in Figure 5
Line 295: staining (not capital letter „S”)
Line 296-297: ..collagen type 1 in the… („type” was missing)
Line 298: ..collagen type 1 in the… („type” was missing)
Line 298: M2-treated MI group (not „M1 group” , „I” was mistyped as 1)
Line 301: Figure 6: Masson (with capital M should be next to Figure)
Line 302: trichrome (r letter was missing)
VERY IMPORTANT: (probably technical event happened at editing)
Line 265 EMT Line 306 EMT results maybe double shown.
These subchapters with immunofluorescence and Western blot are DOUBLE shown.
Figure 5 and Figure 9 are nearly the same (with the exception of some significances/asterixes on panels 5b and 9b
Please check and omit duplicates
Line 330 labeling (not double l)
Line 338: responses (might be plural, not necessary, as you decide)
Line 339: despite, comma is not needed : Despite many clinical
Line 372: literature reported that
Line 377- 378: collagen type I
Line 381: treatment with
Line 390: macrophages play an essential
-
Author Response
Kindly check the attachment.

Reviewer 4 Report
In the presented manuscript entitled “Role of M1 and M2 myocardial macrophages in promoting proliferation and healing via activating epithelial to mesenchymal transition.”, the authors deepened their knowledge on the role of M1 and M2 macrophages in the prevention of myocardial infarctions, however, before publication in the journal Biomedicines, it is advisable to clarify the meanings, standardize the way of writing and use one English language. In order to ensure the reproducibility of the experiments, full/complete explanations of which specific reagents are used by the authors as standard in their laboratory are recommended. Selected, more detailed notes are listed below:
At lines 28, 40, 44, 46, 54, 338, and 376 authors use English (USA) word … remodeling … , but at lines 18, 24, 357, and 391 is English (British) word … remodelling … . Comment: Authors are requested to standardize the content of the manuscript.
At line 57 is: … addressed. This … , but should be … addressed. This … . Comment: A period and a space is written in bold and should be written in normal font.
At line 66 is: … containing 1% antibiotic (Gibco, USA) … , but details are missing though. Comment: After all, the procedure is standard, but an international journal is read by specialists in various related fields who would like to know what antibiotics are used in this field, especially in this isolated case. Please specify the antibiotic used (catalogue number may also be useful).
At line 69 is correctly … 70-80% … , but at line 73 is wrong … 60 -70 % … , and should be … 60–70% …. Comments: 1) Please remove unwonted spaces. 2) Nowadays between numbers a middle sign “ – “is preferred.
At line 88 is: … Biosciences).. … , but there should only be one period at the end of a sentence.
On line 95 at the end of the sentence there is a bold dot and it should be written in normal type.
Sections 2.3., 2.4., 2.5., and 2.6. are doubled in the lines 78 and 175, 89 and 181, 105 and 192, 129 and 204, respectively. Comment: Please renumber.
.
At lines 67, 143, and 153 is … CO2 … , but should be … CO2 … . Comment: The number of atoms of the element should be entered in the subscript, see line such 155.
Also, lines 72, 73, 97 and 98 have ... mL ... . At lines such 82, 85, 167, and 194 it says ... ml ... . Comment: Please codify/standardize in the body of the manuscript.
At line 76 is: … 20ng … , but should be better … 20 ng … . Comment: Please add a space after the number. See line 93. A similar error to correct is on line 74.
On line 80, please provide for the reader from related fields the full name of the EDTA cutter, or the catalog number, preferably the CAS number
At line 93 is: … μg/ mL … , but should be … μg/ml … . Comment: Please check the manuscript sent to the editor and remove unnecessary spaces if necessary.
At line 95 is … 100 μM hydrogen peroxide (Sigma) … , however, there is no information about what hydrogen peroxide, aqueous solution or more stable complex with urea is, and there is no concentration of the aqueous solution used, as I presume. Please add a relevant mention.
In line 115 it is not clear whether this non-fat dry milk contains 5% fat or was dissolved in 19 parts by weight of water. Please make the written information more precise in order to prevent guessing, straight misrepresentation.
At line 169 is … 4-0 … , but should be … 4–0 … .
At line 202 is … 50-100 … , but should be … 50–100 … .
At line 230 is correct … H2O2 … , but at line 226 number of element please writ at subscript. Also please check and correct if necessary at line 434 (at ref. 9).
On line 320 at the end of the paragraph there is a bold dot and it should be written in normal style. Comment: Please correct.
At lines 62, 93, 346 is correctly … In vitro … with Italic style, but at lines 225 and 391 is wrongly … in-vitro … or … in-vitro … , respectively, with an unnecessarily inserted join character “ - “. Comment: Please standardize style at lines 225 and 391.
After chapter 2.9. reappear, new chapters 2.3.–2.6. . Comment: Please renumber the chapters.
Line 388. The conclusions section could be written in more detail.
Lines 404–477. The writing style of references, including the font used, should at least be standardized.
The writing style of the references, including the font used, should at least be standardized and more up-to-date source literature should be added, preferably expanding the introduction.
Author Response
Kindly check the attachment.

Round 2
Reviewer 1 Report
Thank you for addressing some of the issues raised in my initial review.
Please, see again my major points from the initial review. Most of the problems are not addressed in the revised version of the MS.
The main flaw of this MS is insufficient description of methods and very limited description of statistical analysis.
Major points (numbering kept from the initial review):
1. AGAIN: Please, clarify in the MS how many animals were used for experiments, how many are included in each group, how many time points of tissue collection there are (and at what period?). This should be also included in the figures (how many rats and samples are used for each evaluation).
Additionally, please, provide information how animals were killed at the end of experiments.
2. Please, clarify in the MS. For imaging data – clarify if this is from a single section/slide, or a representative image of a larger cohort of images. Please, clarify, which parts of the left ventricle were used for evaluation (MI-free? Scar tissue? Border between scar and normal myocardium?), at what time point was the evaluation done? How many hearts were evaluated?
3. Still to be provided. Please, specify catalogue numbers of antibodies used in experiments (this could be provided either as a separate table or indicated next to the manufacturer of antibodies).
5. Please, provide details on the in vivo model. The in vivo model of MI and macrophage treatment is not clear. Were M1/M2 macrophages administered into the MI-rats? Please, clarify this protocol and provide specific time points that were used for evaluation of the cardiac tissue.
6. The description of statistics is lacking. In the methods authors provide information that mean and SEM are presented, in the Results they provide mean and SD. It is not clear if all data fulfill criteria for parametric tests. Please, confirm that all data fulfilled ANOVA criteria (normality, sphericity, etc.). For data that does not fulfil ANOVA requirements, corresponding to ANOVA non-parametric tests should be carried out. Such data should be also presented as median value with interquartile range (IQR).
7. Please, comment on this - Results of macrophages differentiation and Figure 1 – is there any statistical analysis of the results? Figure 1 is not very convincing that macrophages are clearly differentiated into two different populations. I cannot see a clear-cut difference between M1/M2/M0 macrophages (unless by checking the percentage values). Statistical analysis or CD86 vs CD206 plot could be more convincing?
9. Results – Fig.10 – it seems that there is no apoptosis in the MI group without any treatment in the version in the MS. The same figure in the supp.material is better (but still quite "empty" and dark). Is it possible to improve Fig 10 in the MS? In addition, please, confirm how many sections were evaluated, which parts of the left ventricle were used for evaluation (MI-free? Scar tissue? Border between scar and normal myocardium?), at what time point was the evaluation done? How many hearts were evaluated?
10. There are more original experimental papers that show involvement of macrophages in cardiac remodelling after MI. I strongly suggest including a broader range of original papers, and not to rely mostly on review papers. Please, provide at least a comment from the authors why they would not include any of the suggested recent papers in the revised MS.
Specifically:
Important relatively recent review in the field:
Peet C, Ivetic A, Bromage DI, Shah AM. Cardiac monocytes and macrophages after myocardial infarction. Cardiovasc Res. 2020 May 1;116(6):1101-1112. doi: 10.1093/cvr/cvz336. PMID: 31841135; PMCID: PMC7177720.
The paper showing the role of regulatory T cells in differentiation of macrophages post-MI:
Weirather J, Hofmann UD, Beyersdorf N, Ramos GC, Vogel B, Frey A, Ertl G, Kerkau T, Frantz S. Foxp3+ CD4+ T cells improve healing after myocardial infarction by modulating monocyte/macrophage differentiation. Circ Res. 2014 Jun 20;115(1):55-67. doi: 10.1161/CIRCRESAHA.115.303895. Epub 2014 Apr 30. PMID: 24786398.
The paper showing that shift from M1 to M2 improves healing after MI:
Liu D, Guo M, Zhou P, Xiao J, Ji X. TSLP promote M2 macrophages polarization and cardiac healing after myocardial infarction. Biochem Biophys Res Commun. 2019 Aug 20;516(2):437-444. doi: 10.1016/j.bbrc.2019.06.041. Epub 2019 Jun 19. PMID: 31227217.
The paper showing the effect of CD226 on macrophage polarization and healing of MI heart:
Li J, Song Y, Jin JY, Li GH, Guo YZ, Yi HY, Zhang JR, Lu YJ, Zhang JL, Li CY, Gao C, Yang L, Fu F, Chen FL, Zhang SM, Jia M, Zheng GX, Pei JM, Chen LH. CD226 deletion improves post-infarction healing via modulating macrophage polarization in mice. Theranostics. 2020 Jan 20;10(5):2422-2435. doi: 10.7150/thno.37106. PMID: 32104514; PMCID: PMC7019150.
(please, note I have not authored any of the above papers, I am not associated with any of the authors nor with any of the journals the papers were published in).
Minor points:
1. Again, the MS will benefit from careful proofreading and some corrections of typographical/grammatical errors. Still, some errors have not been corrected. And there are more in the MS to be found and corrected by the authors (and not the reviewer!)
“the mechanism of M1 and M2 myocardial macrophages mediated upregulation of epithelial to mesenchymal transition in promoting proliferation and healing to myocardial have not been fully addressed” - please, clarify “healing to myocardial…” - this is still present in the MS - please, note, this should have been thoroughly checked in the entire MS before submission of the revised version!
MS is generally fine. However, there are some errors that should be corrected before publication.
Author Response
Dear Editor(s) and Reviewer(s), please see the attachment.

Reviewer 2 Report
The authors responded to my comments and made corrections to the text. I have no other comments.
No comments
Author Response

(The authors gave the same response as above.)
